# Policy Implications of the COVID-19 Pandemic on Food Insecurity in Rural America: Evidence from Appalachia

**DOI:** 10.3390/ijerph182312792

**Published:** 2021-12-04

**Authors:** Kathryn M. Cardarelli, Emily DeWitt, Rachel Gillespie, Rachel H. Graham, Heather Norman-Burgdolf, Janet T. Mullins

**Affiliations:** 1Department of Health, Behavior & Society, College of Public Health, University of Kentucky, Lexington, KY 40506, USA; 2Department of Family and Consumer Sciences Extension, College of Agriculture, Food & Environment, University of Kentucky, Lexington, KY 40506, USA; emily.dewitt@uky.edu (E.D.); rachel.gillespie@uky.edu (R.G.); 3Department of Health Management and Policy, College of Public Health, University of Kentucky, Lexington, KY 40506, USA; rachel.hogg@uky.edu; 4Department of Dietetics and Human Nutrition, College of Agriculture, Food & Environment, University of Kentucky, Lexington, KY 40506, USA; heather.norman@uky.edu (H.N.-B.); janet.mullins@uky.edu (J.T.M.)

**Keywords:** rural, food insecurity, COVID-19, food access, social determinants of health

## Abstract

Rural communities are disproportionally affected by food insecurity, making them vulnerable to the consequences of supply disruptions caused by the COVID-19 pandemic. While access to food was initially diminished due to food supply disruptions, little is known about the mechanisms through which federal emergency assistance programs impacted food access in rural populations. Through a series of five focus groups in spring 2021, we examined the impact of the COVID-19 pandemic on food access in a rural Appalachian community in Kentucky. Data were analyzed using a Grounded Theory Approach. Findings revealed the following four primary themes: food scarcity in grocery stores; expanded federal food assistance; expanded community food resources; and expanded home gardening. Participants provided details regarding the way increased federal assistance, especially expanded benefits within the Supplemental Nutrition Assistance Program, allowed them to purchase greater quantities of nutritious food. This study unveils the specific impacts of the COVID-19 pandemic on one rural population, including the influence of some social determinants of health on food insecurity. Policymakers and stakeholders should recognize the layered protection of multiple federal emergency assistance programs against food insecurity and the potential for long-term population health promotion in rural areas.

## 1. Introduction

The COVID-19 pandemic disrupted many aspects of life, including access to food early in the pandemic in the United States [U.S.] [1,2]. Rural and geographically isolated communities are particularly vulnerable to these disruptions, given the existing challenges to wellbeing, including food insecurity—defined by the U.S. Department of Agriculture as “limited or uncertain access to adequate food” [3]—and poorer health relative to their urban counterparts [4]. More specifically, rural Appalachia has experienced multiple challenges to health equity and reports some of the highest levels of chronic-disease morbidity and mortality in the U.S. [5]. Unmet social needs, such as food insecurity, have been linked to poor health outcomes [6,7] and may be a driver of persistent inequities between rural and urban communities [8].

Currently, literature on the impact of the COVID-19 pandemic on food insecurity is emerging. While initial studies highlighted increased food insecurity at the start of the pandemic [1,2], less is known about the status of food insecurity and its relationship with federal relief programs during the pandemic [9]. A study that examined food insecurity patterns in the U.S. early in the COVID-19 pandemic found that Supplemental Nutrition Assistance Program (SNAP) recipients reported less food insecurity compared to those who do not receive SNAP [10]. Two recent studies conducted during the COVID-19 pandemic found reduced food insecurity [11] and fewer unmet health-related social needs among unemployment insurance recipients [12]. Research on the impact of Federal Pandemic Unemployment Compensation found an association between improved food security and receipt of unemployment insurance, although the benefits accrued decreased as the level of assistance reduced [11,13]. However, these studies were national in scope and limited their focus to only one pandemic relief benefit. They did not examine differences based on geography, nor did they explore the specific mechanisms by which assistance might improve food security.

The purpose of this study was to examine the impact of the COVID-19 pandemic on food access in a rural, Appalachian community. We also examined the impact of pandemic-related federal emergency assistance on food access and diet quality. This study provides an early qualitative examination of the impact of these policies on food access in a rural U.S. environment and is particularly timely, given the recent announcement of the permanent expansion of SNAP benefits [14]. Additionally, using qualitative methods to examine food-insecurity experiences of individuals living in a rural area during the pandemic might better equip stakeholders to meet the food and nutrition needs of populations disproportionately impacted by food insecurity. 

## 2. Materials and Methods

### 2.1. Setting

In May and June 2021, focus groups were conducted on food access and physical activity in Martin County, located in eastern Kentucky on the border of West Virginia. Approximately 34% of Martin County residents live in poverty, which is more than triple the national average of 10.5% [15], with 1 in 5 persons considered food insecure [16]. The county experienced a population decline of 13.4% from April 2010 to July 2019 [15], and 9.7% of residents reported unemployment as of July 2021 [17]. The U.S. Bureau of Labor Statistics defines and measures unemployment as persons who found no employment during a reference week and were available for work, excluding persons experiencing temporary illness or leave, and had made efforts to seek some form of employment during a four-week period ending with the reference week [17]. Furthermore, due to the socioeconomic conditions, Martin County is deemed “highly vulnerable” as per the Center for Disease Control and Prevention’s (CDC) Social Vulnerability Index [18].

### 2.2. Recruitment and Eligibility Criteria

In the spring of 2021, our research team purposively recruited adults from the community to participate in focus groups. The Martin County Cooperative Extension Office and Martin County Health Coalition assisted with our recruitment efforts. Informational flyers were placed in the local newspaper and shared via the Martin County Extension Office Facebook page as well as a county Facebook page managed by a Health Coalition member. Eligibility criteria for participation included being 21 years old or older, having resided in Martin County for more than one year, and the ability to be physically active. All participants signed a written consent form and completed a brief sociodemographic survey that included age, race, gender, educational attainment, income, household size, and nutrition-assistance use prior to participating. Focus groups included 10–14 participants per session. Participants were provided with a USD $40 gift incentive for participation.

### 2.3. Measures

The validated six-item food security module of the United States Department of Agriculture (USDA) was used to discern food security status [19,20]. Questions addressed the amount of food eaten within the last 12 months and the ability to afford food within the same timeframe. Scales were used to calculate food security classifications as high or marginal, low, or very low food security and responses were scored using the validated scoring system to discern food security status. Responses of “often” or “sometimes” on questions one and two, and “yes” on questions three through six qualify as an affirmative. Food security status was defined and reflected as follows, based on the sum of affirmative responses: 0–1 high or marginal food security, 2–4 low food security, and 5–6 very low food security. Participants in the low and very low food security groups were classified as food insecure.

A trained moderator facilitated each focus group (K.M.C) using a written semi-structured moderator guide (Appendix A), and two team members took notes during the conversations and monitored the digital audio recorders (E.D. and R.G.). All focus groups were conducted at the Martin County Extension Office and lasted approximately one hour. The University of Kentucky Institutional Review Board approved all study procedures and materials.

### 2.4. Analysis

The focus group conversations were audio recorded and transcribed verbatim by trained researchers and graduate students. Multiple investigators reviewed the transcripts using a Grounded Theory Approach [21] to establish patterns in the data. An iterative inductive–deductive approach was employed by the investigators to identify emergent themes across all five conversations. Investigators discussed the emergent themes for clarity and accuracy to interpret the relevance of patterns observed in the data regarding the impact of the COVID-19 pandemic on food access and diet quality. The themes formed the basis of codes that were analyzed using NVivo software (QSR International, Cambridge, MA, USA, v. 12). All analyses were conducted in May–June 2021.

## 3. Results

### 3.1. Sample Demographic Characteristics

Table 1 provides the sociodemographic characteristics of participants. Fifty-nine adults comprised the final sample across five focus groups. The median age of participants was 57 years, and most were women (*n* = 44, 75%). All participants were White adults, with most participants reporting some college education or were college graduates. The majority of participants reported an annual household income of less than USD $50,000. Participants accurately reflected the sociodemographic composition of the general Martin County population, aside from the fact the majority of participants in the study were female. However, as women are generally the provisioners and preparers of food for their households [22], this difference in representation was deemed favorable to assess the research questions. Furthermore, a household size of 1–2 people was most-often reported among participants (*n* = 12 and *n* = 19, respectively), although 20% of participants indicated household sizes of more than four individuals (*n* = 12).

### 3.2. Food Security Status

Among the reported food security levels, 32.2% experienced low or very low food security and 67.8% reported high or marginal levels of food security. Table 2 provides five additional food insecurity characteristics of our participants using the USDA Household Food Security Survey Module: Six-Item Short Form [19]. The sixth characteristic, eating less or skipping meals, was infrequently recorded, with only six participants reporting their use of this coping strategy every month or in some months.

### 3.3. Qualitative Findings

Four primary themes explaining the impact of the COVID-19 pandemic on food access emerged from the interviews: food scarcity and supply chain disruptions, expanded federal food assistance, expanded food resources, and home gardening. These themes were noted frequently across all five focus groups.

#### 3.3.1. Emergent Theme: Food Scarcity and Supply Chain Disruptions

Participants the frustration and fear of food supply disruptions due to COVID-19, which resulted in a lack of food in grocery stores or other local food outlets. These disruptions led to consumers altering their purchasing patterns. For example, participants shared that scarcity prompted residents to purchase shelf-stable food and, when available, bulk purchase more food than usual during food shopping trips. In addition, participants reported engaging in food-preservation tactics, such as canning, to ensure access to food in the future and to ensure that they were less reliant on the unstable food supply within the community. One participant shared, “A lot of people are doing the storing up their food because of, because of the pandemic. ‘Cause they’re afraid that they’re going to run out of food. And a lot of people will do the canning part so they can reserve their food.”

#### 3.3.2. Emergent Theme: Expanded Federal Food Assistance

For our sample, extensive federal emergency assistance provided ample financial resources and access points for food, including food drop points, free breakfast and lunch for school-aged children through the expanded Summer Feeding Program, and improved economic stability through stimulus payments. Several examples were provided as to how increased SNAP benefits facilitated the purchasing of healthier food and greater quantities of food. For example, one participant noted, “I draw $16 a month before that, and now I get $200, and Lord has that helped me. That’s put food in my deep freezer and on the shelves. And I eat good now!” In addition to the increased affordability of foods, participants shared that they developed their own mechanisms to preserve food using food preservation methods (e.g., freezing), afforded to them through expanded SNAP benefits.

Other forms of federal assistance, such as expanded unemployment insurance, pandemic electronic benefit transfers for participants with dependent school-aged children, and economic impact payments, provided additional income that, in many cases, fostered economic stability. As a result, several participants indicated they experienced levels of economic stability and food security higher than before the pandemic.

#### 3.3.3. Emergent Theme: Expanded Community Food Resources

When participants were asked about how the COVID-19 pandemic affected their access to healthy food, they described access to multiple sources of emergency food assistance that was either expanded or provided a new source of food in the community. These food access points, included existing food pantries and meal service programs, as well as new food access opportunities from charitable organizations, local churches, and meals provided by community members in response to COVID-19.

The local senior citizens’ center was identified as a key resource for nutritious meals in the community, with an increase in service due to COVID-19. One participant explained, “At the senior citizens’ center, we picked up, uh, I think 15 extra on home delivered meals, and then we picked up 10–15 on curbside meals they came and picked it up. Additionally, then the ones that were way up the county line, they started sending them frozen meals. They called them mom meals.” Further, critical food access points were flexible in their capacity to provide food to individuals in need as demonstrated by this quote.

Collectively, these examples of expanded and newly created food-access points demonstrate the drive within the community to meet the needs of their residents. Many participants described an increase in services provided by community organizations in the context of the community value of aiding one another during times of crisis or during emergency.

#### 3.3.4. Emergent Theme: Expanded Home Gardening

The provision of additional time to plant and maintain a home garden was described by participants as a common COVID-19 pandemic activity. One participant shared, “There’s a lot of people that garden around here. I know I’ve seen gardens everywhere.” Participants also mentioned the fear of food scarcity as a reason for increased gardening and that a motivation for gardening may include sharing one’s home garden harvest with family and friends. Furthermore, the availability of fresh produce from gardening provided an opportunity to use food preservation tactics to reserve food for the future or to sustain food supplies through winter.

Table 3 provides a summary of themes and representative quotations that denote these themes.

## 4. Discussion

The experiences of participants in this rural, Appalachian community demonstrated that, although the COVID-19 pandemic disrupted food supplies and initially diminished food access in grocery stores, the expansion of other food access opportunities and resources, such as home gardens, existing and pop-up food pantries and meal distribution systems, more than accounted for any food access shortages. Community-led efforts to expand food access, including sharing home garden harvests and creating new emergency food access points, are consistent with previous findings in this community that noted a strong cultural value for assisting one another [23]. This is similar to the study of Hege et al. [24], in which they noted a strong sense of social cohesion and social capital in rural communities, which contributes to a higher likelihood of community members assisting one another in times of need. In addition, participant responses indicated a strong cross-sector response to food needs with local community-based organizations providing aid.

While our sample indicated the ability to procure food of better nutritional quality, it is important to note that 32% of our study participants still reported either low or very low food security, compared to the overall statistic of 11% of U.S. households in 2020 [9]. The impact of the COVID-19 pandemic on food security has been observed in similar low-income populations. Qualitative findings from eastern Tennessee reveal additional barriers to food security, including increased food prices, quantity restrictions, and a limited stock of grocery items [25]. Furthermore, Mui et al. [26] found that food-insecure adults in urban areas were negatively impacted by changes in employment status and limited availability of culturally preferred foods. Coupled with observations from this rural Appalachian sample, these findings demonstrate food insecurity to be a complex public health challenge compounded by many influential factors. In particular, access issues and income restraints in these populations makes it difficult to procure the necessary food provisions. Multiple approaches from communities and federal agencies provide an opportunity to intervene.

For the past several decades, federal agencies have sought to not only reduce food insecurity but also bolster nutrition security in resource-constrained households. While the primary aim of these programs is to allow consistent access, availability, and affordability of nutritious foods by providing benefits and financial resources to these families, these efforts may not consider the host of factors affecting the procurement of food, and states still report difficulties in managing food insecurity among their residents. Kentucky is one of nine states in which prevalence of food insecurity (13.8%) is significantly higher than the national average (10.7%), yet Kentucky experienced a non-significant decline in food insecurity for 2018–2020 compared to the two preceding decades [9]. In a statewide survey in Vermont, the impact of the COVID-19 pandemic on food security was assessed using language associated with the year before, or since, the outbreak [1]. Among this primarily rural population, an increase of almost one-third in food insecurity was reported for March-April 2020, prior to the distribution of federal stimulus funds. From these two examples, we can see that, regardless of federal efforts, food insecurity was increasing or not improving for a variety of reasons prior to the distribution of federal stimulus funds as a result of the COVID-19 pandemic. Taken together, these findings suggest that federal emergency assistance efforts provided an important safety net for rural families during the pandemic.

As observed by McElrone et al. [25], both food specific and general econominc resources sustained participants and helped meet the needs of their families during the COVID-19 pandemic. Results from our study reinforce findings from earlier quantitative studies noting reductions in food insecurity among Americans receiving unemployment insurance benefits [11]. Raifman et al. [11] reported a reduction in food insecurity with the receipt of a federal stimulus payment, although these associations were not statistically significant. Similarly, we found that participants perceived a reduction in food insecurity associated with federal stimulus payment or SNAP benefits. Our findings highlight, in detail, the mechanisms that helped improve access to food in this Appalachian population. Participants provided specific examples of the extent to which their SNAP benefits increased and the way additional assistance from stimulus payments allowed them to purchase more food and food of higher nutritional value. They also detailed the importance of free breakfast and lunch for children through the Summer Feeding Program and the additional availability of food through community programs.

Given a larger percentage of households in rural communities participate in SNAP than in urban areas as a percentage of state populations [8], it is likely that policies seeking to improve food security may help reduce the persistent inequities between rural and urban communities. Our findings provide evidence of the potential impact of such assistance on food access in an Appalachian community. Southern rural community members, regardless of food security status, have been found to use community resources such as cooking classes, farmers’ markets, and community gardens equally [27,28]. Furthermore, SNAP benefits provide a rapid, effective economic stimulus, generating USD $1.70 for each dollar spent [29].

Federal efforts to aid Americans during the pandemic included pandemic electronic benefit transfer (EBT), expanded unemployment insurance, expanded SNAP benefits, expanded child tax credit and earned income credit, emergency rental assistance, and three rounds of economic impact payments [30]. Our results suggest that federal emergency assistance helped to improve individual economic conditions, with focus group participants reporting greater economic stability than before the pandemic. This finding is particularly relevant considering the growing emphasis on addressing the social determinants of health, which comprise the conditions in which people work, live and age [31], including employment, the built environment, social isolation, and the education system [32]. Developing policies and programs that provide financial support that extends beyond the COVID-19 pandemic may be an effective mechanism to address unmet social needs and reduce poor health outcomes. Multi-level approaches can directly address the social determinants of health, driving food insecurity as a result of poor economic conditions [33]. Given the robust evidence linking income and health [34], policies that include the observed elements listed above may have wide-ranging health effects. For example, policies that provide supplemental income [35,36,37] and reduce housing instability [38,39,40] have been shown to improve health outcomes. In addition, increasing access to food through policy-based efforts not only improves nutritional status but has far-reaching implications on overall health and wellbeing, especially in health-disparate, rural communities [24].

As Hege and colleagues stressed [24], the priority should be in addressing the systemic causes of health disparities within rural communities, creating a ripple effect of positive systems change. The intersection of tangible action and systems change lies in policy, which addresses multiple layers of complex public health problems [41]. Policy makers should recognize the importance of multiple federal emergency assistance programs on food access and the potential for long-term population health promotion. For example, the impact of SNAP benefits on poverty, food insecurity, and health are well documented [42]. On August 16, 2021, USDA announced the first update to SNAP benefits in 45 years to reflect contemporary food costs [14]. SNAP participants recognize the program as one factor in the larger effort required to address rural poverty and health disparities [43]. The newly revised SNAP benefits will serve as a natural experiment to monitor food security and diet quality looking forward. Additional complementary strategies provide a unique opportunity for substantial impacts. Online SNAP redemptions became widely available via a pilot program during the COVID-19 pandemic with positive results [44]. The permanent expansion of benefits could increase program participation, alleviate the burden of stigma associated with SNAP, and protect populations from food insecurity. The increased flexibility for SNAP participation, including policies to expand online benefit redemption, warrant additional exploration.

### Limitations

This study had several limitations. Our study sample was not randomly selected. We employed a purposive sampling approach, and our findings may not be generalizable to Martin County and certainly not to all rural populations. Our sample age is older, had more females and higher education relative to county population estimates, which may suggest that the barriers to food access were potentially underestimated in our findings. Additionally, it is possible that participants’ comments were tempered by a social desirability bias. Future research should investigate whether these experiences are similar in other rural areas. Additionally, future studies should examine whether some of the food access mechanisms that investigators observed in this rural community could be translated into addressing other social determinants of health inequities. It is yet to be determined if unique properties of rural communities allow them to negotiate food insecurity challenges differently than their urban counterparts.

## 5. Conclusions

This study identifies the specific impact of the COVID-19 pandemic on one rural population and highlights some of the social determinants of health on food insecurity. Although our sample was limited to one rural Appalachian community, it is likely that improved food security resulting from federal emergency assistance, as demonstrated by our findings, extends to other rural and socioeconomically disadvantaged communities. Policymakers and stakeholders should recognize the layered protection that multiple federal emergency assistance programs provide for individuals living in rural areas, and the potential for long-term population health promotion, by addressing food insecurity in these areas.

## Figures and Tables

**Table 1 ijerph-18-12792-t001:** Sociodemographic characteristics of focus group participants (*n* = 59) and of Martin County, Kentucky residents, 2021.

Characteristic	Among All Participants *n* (%)	Martin County, KY Residents ^5^ %
Age (median)	57 years	39 years
**Gender**		
Female	44 (75%)	45%
Male	15 (25%)	55%
**Race**		
White	59 (100%)	92%
**Education**		
11th grade and below	6 (10%)	26%
High school graduate or GED	24 (41%)	39%
Some college	12 (20%)	25%
College graduate	17 (29%)	10%
**Household Income (USD)** ^1,2^		
<$20,000	21 (36%)	
$20,001–$49,999	20 (34%)	
>$50,000	16 (30%)	
**Nutrition Assistance** ^1,2,3^		
SNAP	15 (25%)	
Food pantries	8 (14%)	
Other ^4^	2 (3%)	
No assistance	29 (49%)	

^1^ Some participants chose not to respond. ^2^ No analogous data categories are available from the U.S. Census Bureau. ^3^ Participants could select more than one category of nutrition assistance. ^4^ Other forms of nutrition assistance include Senior Farmers Market Nutrition Program Vouchers and Medicaid. ^5^ Data from the U.S. Census Bureau. Abbreviations: KY: Kentucky; GED: Tests of Graduate Educational Development; USD: United States Dollar; SNAP: Supplemental Nutrition Assistance Program.

**Table 2 ijerph-18-12792-t002:** Participant food insecurity characteristics of focus group participants (*n* = 59).

Question	Often or Sometimes True *n* (%)	Never True *n* (%)
The food I bought just didn’t last and I didn’t have money to get more ^1^	21 (36%)	37 (63%)
I couldn’t afford to eat balanced meals ^1^	29 (49%)	29 (49%)
	**Yes *n* (%)**	**No *n* (%)**
In the last 12 months, did you or other adults in your household ever cut the size of your meals or skip meals because there wasn’t enough money for food? ^1^	8 (14%)	50 (85%)
In the last 12 months, did you ever eat less than you felt you should because there wasn’t enough money for food?	8 (14%)	51 (86%)
In the last 12 months, were you ever hungry but didn’t eat because there wasn’t enough money for food? ^2^	7 (12%)	51 (86%)

^1^ Some participants indicated “I don’t know”. ^2^ Some participants chose not to answer.

**Table 3 ijerph-18-12792-t003:** Themes and representative quotations from focus groups in Martin County, Kentucky, 2021.

Themes	Representative Quotations
**Food scarcity and supply chain disruptions**	“I think people were scared they weren’t going to have nothing to eat. They said there was going to be a shortage… On everything there’s a shortage cause they can’t get truck drivers to bring it in. And I think people were raising potatoes and stuff that they can store, you know what I mean. Canned beans and other stuff…. I think it’s easing off, but you know, at one time it was, you know, people worried. You could go to the Dollar General and the shelves were empty.” “We went to Wal-Mart and there wasn’t anything there. You know.” “It was spooky too. I thought walking around and seeing nothing on the shelves…” “Shelves were empty.” “Scary, scary.” “I thought oh my, this could be real.’” “I think it’s easing off but you know, at one time it was, you know, people worried. You could go to the Dollar General and the shelves were empty.” “I think it’s something that’s probably gonna persist for a little while. I mean, you know, because where they have had a shortage, everybody’s wanting to get it, or, they can’t get it, like fertilizer, you can’t get that, wood, you can’t get that, you can’t, it seems like you can’t get anything, or if you do it’s very limited.”
**Expanded federal food assistance**	“And talking about the food stamps for this pandemic, I was getting like a hundred fifty something, and then they cut me down to $16. And then this pandemic, you know come up, and then I started going out to the foodbanks to help get food for me and my granddaughter and, with me and her together, I get like $1500 or something to live on. By the time I pay my bills, I don’t have much money to get out and buy food with. And this pandemic helped me in ways uh, ‘cause it give me like, between both of us, $400 in stamps. ” “All the food stamps, all the, you know, the kids getting meals delivered. I, I don’t think anybody’s ever run out of food.” “But there was also a lot of EBT *. They gave out extra stamps and stuff like that. So, my guess would be they probably ate better food because they had more money to buy it with, better stuff. I know I did because I got a little extra. I only got $6 (laughs). That’s all I used to get, and they gave you a little bit more. I can buy more fruits and vegetables.” “The number of families [relying upon pantries] has gone way down. I think, because they get um they’re getting uh so they’re getting, some people are getting extra food stamps, if they get food stamps. Then their child, regardless of if they get food stamps or not, is getting a food stamp card, and then if they’re in school they’re getting food. If they’re not in school, the bus is coming to our house to bring them the food. It’s just getting so much so much everywhere.”
**Expanded community food resources**	[From food pantry employee] “Our numbers have dropped… I don’t know the reasons, I’m trying to figure it out. Our numbers have consistently dropped the last probably year. We were doing about maybe around a 100 a month, 100 families. Now we’re down in the 50s… And we’ve talked, we’ve talked to other counties as well their numbers have gone down. So, I think it’s like, and this is my personal opinion, all the money that’s been given out, people just they don’t have to go to the food pantry I guess.” “The amount of food banks has increased, the possibilities of food have increased. I wouldn’t think there would be too many, I would hope not.” “We had a rush of like emergency people that never came back after the first month that they got it um I don’t know if it was a little bit of a panic that the grocery store would be out of the food or something or they just wanted to see what food we offered and then they’re like uh I don’t really want that or something I don’t know. I think the ones that are the most needy have stayed because they really do rely on it.”
**Expanded home gardening**	“I feel that there are a lot of farmers, but they garden for their own personal use. I know that a lot of people do canning and have the cellars to store and that type of thing. I know that the Extension office has worked on a program, and I think they doubled last year.” “I think a lot of people do the garden so that they can help other people. Uh, a lot of people will take their extra produce that they do get out of it and give it out to their families to help them along the journey of hardship and stuff. So, instead of throwing it away they’d rather give it away.” “I think a lot of people around here have a garden even if it’s just a 6 × 6 plot in their yard with just enough to get them through the winter.” “I know a lot of people that really kicked it up a notch in the gardening, and you know, ‘cause it happened you know about the time that everybody were putting things you know planting. And um, I know a lot of people that said ‘okay, this could be serious I’m going to really take this garden serious’, and ‘I’m going to really put everything away that I can’. And they share too, but I’m just saying I just think that a lot of people kind of kicked it up a notch, I know I did you know, I put away everything that I could that you know, probably more though, probably more than I had ever done. Just if, you know, I think a lot people that could, did that.”

* EBT: electronic benefit transfer.

## Data Availability

No appliable.

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
