# Peer review of "Policy Implications of the COVID-19 Pandemic on Food Insecurity in Rural America: Evidence from Appalachia"

_ijerph, 2021, doi:10.3390/ijerph182312792_

Round 1

Reviewer 1 Report

The manuscript was well written and very relevant. However, revisions are needed as follows:

Materials and methods:

- It would be good to add that a mixed methods design was followed - a lot of emphasis is placed on the qualitative aspect, but quantitative results were also reported

- The number of FGDs were reported as a result, but it would have been good if the number of FGDs and how many participants were in each of the FGDs could be reported in the methods

- Mention is made of the semi-structure moderator guide - could you include the questions/statements that were included for the FGDs

- On page 2, line 79: Write out CDC before the abbreviation is used

- The sampling should be better explained - in the results the total sample is reflected as 57 participants and 39 from Martin County. Where did the other 57 participants come from? Or were the 39 from Martin County included in the 57? If so, where did the others come from and why were they included if the study was only conducted in Martin County? Martin County is described as poverty-stricken and it is understood why it is in included in the sampling, but nothing is said about the other participants . This is confusing as e results in Table 2 include all the participants and then in Table 3 it seems as if it only included the results from the participants from Martin County. Why were the results in table 2 not separated as in table 1?

Results

- Page 2 lines 122-123: Most is the largest % whereas the majority reflects >-50% of the respondents. The way the results are reported here should be revised as most had college education and the majority had household income of <- $50,000

- Page 4 line 133-134 should be re-phrased: "Among..., 32.2% experienced low or very low and 67.8% high ....security respectively".

Discussion

- Good and relevant

- Page 8 line 236: EBT not abbreviated before

Conclusions

I am not sure if you can conclude that the study highlights the social determinants of health on food insecurity. Few social determinants of health were measured, eg. economic stability, neighborhood and physical environment, health care system, community conditions and safety, lifestyle factors, living and working conditions, etc were not measured. No correlations were done between education, household income and gender with food security for eg. Thus no associations determined.  The first sentence should thus be re-phrased.   

Author Response

Thank you so much for your feedback on our manuscript. Please see below for an itemized response to each point.

Reviewer Comment

Author Responses/Revisions

It would be good to add that a mixed methods design was followed - a lot of emphasis is placed on the qualitative aspect, but quantitative results were also reported

While we did compile participants’ socioeconomic characteristics and food security status (both of which included quantitative measures), our methods were primarily qualitative. Instead of identifying our study as a qualitative study, in the final paragraph of the introduction, we edited the text to clarify that our study provides a qualitative examination of factors affecting food insecurity.

The number of FGDs were reported as a result, but it would have been good if the number of FGDs and how many participants were in each of the FGDs could be reported in the methods

We added in the methods section the range of numbers of focus group participants per focus group.

Mention is made of the semi-structure moderator guide - could you include the questions/statements that were included for the FGDs

We have included the moderator guide as an Appendix for reference to the readers.

On page 2, line 79: Write out CDC before the abbreviation is used

We have made this correction.

The sampling should be better explained - in the results the total sample is reflected as 57 participants and 39 from Martin County. Where did the other 57 participants come from? Or were the 39 from Martin County included in the 57? If so, where did the others come from and why were they included if the study was only conducted in Martin County? Martin County is described as poverty-stricken and it is understood why it is in included in the sampling, but nothing is said about the other participants . This is confusing as e results in Table 2 include all the participants and then in Table 3 it seems as if it only included the results from the participants from Martin County. Why were the results in table 2 not separated as in table 1?

Table 1 was intended to compare the demographics of our study sample to the overall county demographics. For example, the median age of all focus group participants was 57 years, and the median age of all Martin County residents was 39. We have added years after these numbers in Table 1 to clarify this. Table 2 includes responses from all 59 focus group participants, and we added language in the Table 2 title to clarify this. Comparable food insecurity data are not available for all of Martin County, which is why we did not include that in Table 2.

Page 2 lines 122-123: Most is the largest % whereas the majority reflects >-50% of the respondents. The way the results are reported here should be revised as most had college education and the majority had household income of <- $50,000

Thank you for this- we have clarified this in the text.

Page 4 line 133-134 should be re-phrased: "Among..., 32.2% experienced low or very low and 67.8% high ....security respectively".

We have made this correction in the text.

Page 8 line 236: EBT not abbreviated before

We spelled this out in the text.

I am not sure if you can conclude that the study highlights the social determinants of health on food insecurity. Few social determinants of health were measured, eg. economic stability, neighborhood and physical environment, health care system, community conditions and safety, lifestyle factors, living and working conditions, etc were not measured. No correlations were done between education, household income and gender with food security for eg. Thus no associations determined.  The first sentence should thus be re-phrased.   

We understand this concern, so we have modified that sentence to reflect that we did not seek causal associations or correlations.

Reviewer 2 Report

The article explores a very relevant topic within food insecurity in rural areas, which is the impact of policy actions. The qualitative approach used through focus groups is well-sounded and appropriate, and the paper has the potential to add relevant examples to the literature of food insecurity in rural areas.

The overall concept is interesting and well-designed. However, I feel there are some points that the authors might want to consider in order to reinforce the robustness of the qualitative findings and make them more relevant and easier to follow to non-US readers. The main points I would like the authors to consider are the following:

First, regarding the methodology, I consider the description of the extent and focus of the focus groups could use more detail. An insight into the moderator guide and the contents that were touched upon seems necessary (e.g. themes/questions included in the guide, provocation question, etc.).

Also, more information is needed about the focus groups: how many participants were in each focus group and how the different groups were similar/different. Commenting on how the participants represent/don’t represent well the population, and what are the critical points to consider regarding this would be optimal (For example, is a problem that 75% of the participants were female? Maybe the answer is not because women are still in general, the ones in charge of the management of food provision and diets (acquiring groceries, cooking, etc.)… I feel that a reflection in this vein could support the robustness of the study. The same goes for the sampling unbalances regarding education and race).

For this, the information now under section 3.1 (including the table, or at least the most part of it**see the comment at the end) could be moved to section 2.2. As it is now, the characterisation of the sample is fragmented, and it might confuse the reader a bit.

Second, the report on the qualitative findings looks rather weak at the moment and could be enriched by providing more detail into the emergent themes.

  • It is unclear how/why a theme is considered as ‘emergent’ (e.g. is it because it was repeated in all the focus groups and/or shared among participants with different profiles?). For example, information on the ‘weight’ or importance of the theme in the overall discussion would be appreciated by the reader.
  • Also, enriched details in the description of the themes would be helpful (for instance, including examples). I’m aware that, to some extent, this is the purpose of table #3, but I feel it doesn’t work so well. As a reader, I would appreciate if illustrative quotations were integrated into the description of the themes, or at least after each theme.
  • Regarding the quotations now in table #3, I feel that they could be tidied up and slimmed down, maintaining the messages but removing unnecessary wording (As the analysis didn’t include discourse analysis, I feel it is not relevant to keep discourse sings and the complete verbatim) and unclear quotes (for example quote #5 in theme 1 “I think it’s something that’s probably gonna persist…” is not clear and otherwise requires additional explanation).  In any case, if the authors decide to keep it as it is now, Table #3 containing the quotations from focus groups probably would work better if it was placed after the themes are described. Also, more analytical detail in the table would help to show the relevance of the statements. Including a profile of the type of participant who provided the quote according to the socio-demographic and food insecurity characterisation of the participants would be an excellent addition (regardless of the quotations kept in this table, distributed in several tables by theme, or integrated into the text).

Third, I found myself a bit lost at some points when reading the manuscript because I am not a researcher already familiar with the US food insecurity and policy context regarding this topic. In this regard, there is contextual information in the discussion that would be helpful to be included or introduced in the Introduction section when setting the scene (for example, sentences in lines 2013-209).

Following on the discussion, I feel that some of the content would probably work best if placed in the results section, and so contributing to the more detailed description of the themes. For example, the first sentence of the second paragraph on page 8 (lines 212-214) could go under section 3.3.2. This move would allow us to go deeper in the discussion in some points that at the moment are only mentioned. For instance, in lines 200-201, it is mentioned that food insecurity is compounded by many factors, and as a reader, I feel I want to know more. So, I would suggest the authors to follow up on that sentence with a list of those factors (particularly of the ones that are going to be touched upon later on in the next few paragraphs). Similarly, lines 22-225 remit to mechanisms and examples that have been provided earlier. Including them here again, in a systematic way would reinforce the message and help enormously to the reader to follow the point.

Last, a comment on tables #1 and #2:

  • At the moment, table #1 is labelled as “Demographic characteristics…” but it includes data on household income, nutrition assistance and food security status for the sample that don’t have analogue comparative data for the county. Thus, I would recommend reconsidering the information shown in the table and limiting it to age, gender, race, education (and maybe household income) and, as I mentioned before, moving the table to the description of the sample.
  • Then all the information regarding the food security status of the sample could be included in table #2  (with the columns reporting for low/very low food security and marginal/high food security for the questions and the indicators that are now included in table 1).
  • Thus, a new table reporting only on nutrition assistance could be included in the results section to reinforce and highlight this point which is at the core of the study (maybe including more information or breaking down the data if it is available)

Author Response

Thank you so much for your feedback on our manuscript. We have itemized our responses to your suggestions below.

Reviewer Comment

Author Responses/Revisions

First, regarding the methodology, I consider the description of the extent and focus of the focus groups could use more detail. An insight into the moderator guide and the contents that were touched upon seems necessary (e.g. themes/questions included in the guide, provocation question, etc.).

We added how many participants were in each focus group session in the methods. We have also included the Moderator Guide in the Appendix for reference.

Also, more information is needed about the focus groups: how many participants were in each focus group and how the different groups were similar/different. Commenting on how the participants represent/don’t represent well the population, and what are the critical points to consider regarding this would be optimal (For example, is a problem that 75% of the participants were female? Maybe the answer is not because women are still in general, the ones in charge of the management of food provision and diets (acquiring groceries, cooking, etc.)… I feel that a reflection in this vein could support the robustness of the study. The same goes for the sampling unbalances regarding education and race).

We appreciate this thoughtful feedback and have incorporated further details describing each of the focus groups and their comparison to the general Martin County population. The following language has been added in an effort to contribute and depict the robustness of this study within the context of this specific Appalachian community, “compared with the Martin County population, participants were largely accurately reflected aside from the majority female participation in the study. However, as women are generally the provisioners and preparers of food for their households, this difference in representation was favorable to assess research questions”. Additionally, we added in the methods section the range of numbers of focus group participants per focus group.

For this, the information now under section 3.1 (including the table, or at least the most part of it**see the comment at the end) could be moved to section 2.2. As it is now, the characterisation of the sample is fragmented, and it might confuse the reader a bit.

Thanks for pointing out this confusion. We modified the title of section 3.1 to reflect its contents (Recruitment and Eligibility Criteria). We believe that the description of the study sample in the results section is appropriate.

Second, the report on the qualitative findings looks rather weak at the moment and could be enriched by providing more detail into the emergent themes. It is unclear how/why a theme is considered as ‘emergent’ (e.g. is it because it was repeated in all the focus groups and/or shared among participants with different profiles?). For example, information on the ‘weight’ or importance of the theme in the overall discussion would be appreciated by the reader.

 We appreciate this feedback. Emergent themes were identified based upon patterns in the data across the five focus groups; similarities in patterns were collapsed to form themes. We have incorporated additional detail in sections 2.4 and 3.3 to provide clarity on emergent themes.

Also, enriched details in the description of the themes would be helpful (for instance, including examples). I’m aware that, to some extent, this is the purpose of table #3, but I feel it doesn’t work so well. As a reader, I would appreciate if illustrative quotations were integrated into the description of the themes, or at least after each theme.

Thank you for this comment. The narrative descriptions of each emergent theme have been enriched to include examples as well as short illustrative quotes, in addition to the comprehensive table.

Regarding the quotations now in table #3, I feel that they could be tidied up and slimmed down, maintaining the messages but removing unnecessary wording (As the analysis didn’t include discourse analysis, I feel it is not relevant to keep discourse sings and the complete verbatim) and unclear quotes (for example quote #5 in theme 1 “I think it’s something that’s probably gonna persist…” is not clear and otherwise requires additional explanation).  In any case, if the authors decide to keep it as it is now, Table #3 containing the quotations from focus groups probably would work better if it was placed after the themes are described. Also, more analytical detail in the table would help to show the relevance of the statements. Including a profile of the type of participant who provided the quote according to the socio-demographic and food insecurity characterisation of the participants would be an excellent addition (regardless of the quotations kept in this table, distributed in several tables by theme, or integrated into the text).

Thank you for these suggestions. We have moved Table 3 to after the themes are described. We have also trimmed quote #5 under theme 1.

Third, I found myself a bit lost at some points when reading the manuscript because I am not a researcher already familiar with the US food insecurity and policy context regarding this topic. In this regard, there is contextual information in the discussion that would be helpful to be included or introduced in the Introduction section when setting the scene (for example, sentences in lines 2013-209).

Thank you for the opportunity to clarify our discussion and provide additional context. In the newly revised manuscript, the third paragraph of the discussion sections includes an overview of US efforts to mitigate food insecurity, which sets the stage for the remaining portion of the discussion.

Following on the discussion, I feel that some of the content would probably work best if placed in the results section, and so contributing to the more detailed description of the themes. For example, the first sentence of the second paragraph on page 8 (lines 212-214) could go under section 3.3.2. This move would allow us to go deeper in the discussion in some points that at the moment are only mentioned. For instance, in lines 200-201, it is mentioned that food insecurity is compounded by many factors, and as a reader, I feel I want to know more. So, I would suggest the authors to follow up on that sentence with a list of those factors (particularly of the ones that are going to be touched upon later on in the next few paragraphs). Similarly, lines 22-225 remit to mechanisms and examples that have been provided earlier. Including them here again, in a systematic way would reinforce the message and help enormously to the reader to follow the point.

Thank you for these helpful suggestions. We have made the suggested edits to the current text and further elucidated the highlighted points in the text.

Last, a comment on tables #1 and #2:

At the moment, table #1 is labelled as “Demographic characteristics…” but it includes data on household income, nutrition assistance and food security status for the sample that don’t have analogue comparative data for the county. Thus, I would recommend reconsidering the information shown in the table and limiting it to age, gender, race, education (and maybe household income) and, as I mentioned before, moving the table to the description of the sample.

Then all the information regarding the food security status of the sample could be included in table #2  (with the columns reporting for low/very low food security and marginal/high food security for the questions and the indicators that are now included in table 1).

Thus, a new table reporting only on nutrition assistance could be included in the results section to reinforce and highlight this point which is at the core of the study (maybe including more information or breaking down the data if it is available)

Thank you for these suggestions. We renamed Table 1 so that it now reflects the sociodemographic characteristics of focus group participants. We deleted the food security characteristics at the end of Table 1, as those results were shared in the first sentence of section 3.2. We renamed section 2.2 to Recruitment and Eligibility Criteria to better reflect its contents.

Round 2

Reviewer 2 Report

Congratulations on the paper and probably one of the most efficient addresses of reviewer's suggestions that I've seen this year :D

Author Response

Thank you for your kind comments.